# Impact of cavity on interatomic Coulombic decay

Lorenz S. Cederbaum [1✉] & Alexander I. Kuleff [1✉]

The interatomic Coulombic decay (ICD) is an efficient electronic decay process of systems embedded in environment. In ICD, the excess energy of an excited atom $A$ is efficiently utilized to ionize a neighboring atom $B$. In quantum light, an ensemble of atoms $A$ form polaritonic states which can undergo ICD with $B$. Here we investigate the impact of quantum light on ICD and show that this process is strongly altered compared to classical ICD. The ICD rate depends sensitively on the atomic distribution and orientation of the ensemble. It is stressed that in contrast to superposition states formed by a laser, forming polaritons by a cavity enables to control the emergence and suppression, as well as the efficiency of ICD.

[1] Theoretische Chemie, Physikalisch-Chemisches Institut, Universität Heidelberg, Heidelberg, Germany. ✉email: lorenz.cederbaum@pci.uni-heidelberg.de; alexander.kuleff@pci.uni-heidelberg.de

Interatomic or intermolecular Coulombic decay (ICD) is a nonlocal and efficient electronic decay mechanism taking place in weakly bound matter. ICD becomes operative once the excess energy of an excited atom or molecule suffices to ionize a neighbor[1]. The energy released by the electronic relaxation of this excited atom or molecule ionizes the neighbor and hence energy conservation is fulfilled without the need for nuclear motion. As a consequence, the excited species, as well as the neighbors, can be atoms or molecules and the timescale involved is typically in the femtosecond regime[2–4]. Being ultrafast, ICD quenches in most cases concurrent electronic and nuclear mechanisms[5–8]. ICD has a wide range of applications. It has been shown to be active in the quantum halo systems $He_2$[9,10] and LiHe[11] where the mean separation of the atoms is extreme, in quantum dots and quantum wells[12–15], and its potential importance in radiation damage and for biologically relevant systems have also been discussed[5,6,16–20]. A recent review covers the fundamental and applied aspects of ICD and related processes[21].

The interaction of atoms and molecules with quantized radiation field like that inside a cavity has lead to an active new area of research, which opens up many possibilities to manipulate their properties, to enhance or suppress available mechanisms, and to mediate new ones. Among the long list of possibilities, we mention control of photochemical reactivity[22,23], control of chemical reactions by varying the properties of the quantized field[24–27], enhance charge[28–31], and energy-transfer[30,32] processes, and increase non-adiabatic effects in molecules[26,33,34]. It is known that a classical laser field can induce a conical intersection even in a single diatomic molecule[35–37]. Indeed, a quantized radiation field also induces a conical intersection in a diatomic with new implications on its dynamic properties[38–40] and, of course, also in polyatomics[40,41]. New types of intersections appear when more molecules are subject to the same quantized field where the molecules interact with each other via the field. Here, we mention the collective conical intersection, which gives rise to unusual dynamics[42].

The main aim of the present work is to demonstrate and discuss the substantial impact the interaction with quantized light exerts on ICD. To be specific, we concentrate on atoms. Due to the interaction with the cavity mode, mixed electronic-photonic (polaritonic) states are formed[43]. In polaritonic states the atoms are entangled and one can expect interference effects to play a role. Constructive interference effects have been shown to enhance resonant photoionization in a multiatom ensemble[44,45]. A cavity is a particularly suitable platform to investigate ICD as the entanglement is naturally produced in the polaritonic states. Since the energies of polaritonic states can be manipulated, we shall see that cavities enable opening and closing the ICD channel which is not possible for a superposition state formed by a laser. There is a resemblance between polaritonic states and coherent superposition states of an ensemble formed by a laser, which will be addressed after we have introduced and applied the ICD to polaritonic states. In this first study, we stress the fundamental aspect of the impact of quantum light on ICD, namely that in addition to forming superposition states it enables to control the emergence and suppression, as well as the efficiency of ICD. There are several types of cavities available nowadays, and this will be discussed too.

## Results

We consider an ensemble of $N$ non-interacting identical atoms of the kind $A$ in a cavity with a quantized light mode (cavity mode) of frequency $\omega_c$ and polarization direction $\epsilon_c$. The total Hamiltonian of the ensemble-cavity system reads[43,46–48]:

$$H = H_e + \hbar\omega_c \hat{a}^\dagger \hat{a} + g\, \epsilon_c \cdot \mathbf{d}(\hat{a}^\dagger + \hat{a}), \quad (1)$$

where $H_e = \sum_{i=1}^N H_i$ is the electronic Hamiltonian of the ensemble, $\mathbf{d} = \sum_{i=1}^N \mathbf{d}_i$ is the total dipole operator of the ensemble, $g$ is the coupling strength between the cavity and the atoms, and $\hat{a}^\dagger$ and $\hat{a}$ are creation and annihilation operators for the bosonic electric field mode. The quadratic dipole self-energy term is neglected as it is only of relevance for very strong coupling.

Since all atoms are of the same kind and we assume the cavity mode to be resonant with an excited atom $A^*$, it is straightforward to find the energies and eigenstates of the above Hamiltonian in the single-excitation space. For that purpose, we define the contributing space to be spanned by $\{A_1 A_2 \ldots A_N 1_c\}$, which is the configuration state of the ensemble in its electronic ground state and the cavity in a single-photon state, and the $N$ configuration sates $\{A_1 \ldots A_i^* \ldots A_N 0_c\}$, $i = 1, \ldots, N$, where one atom is excited and the remaining $N-1$ are in their electronic ground state and the cavity has zero photons. Representing the Hamiltonian (1) in a single-excitation space, generally leads to an arrowhead matrix the properties of which have been analyzed in various contexts[49–51]. In the present case, the matrix is particularly simple and can be solved in closed form.

It is well known that one obtains two so-called bright states and $N-1$ dark states. Choosing the energy of $A$ in its ground electronic state to be the origin of the energy scale, the bright states have the energies $\hbar\omega_c \pm \sqrt{N}g$ and the dark states are degenerate with energy $\hbar\omega_c$. The eigenstates of the former are

$$\Phi_{up/lp} = \frac{1}{\sqrt{2}}\left[\{A_1 \ldots A_N 1_c\} \pm \frac{1}{\sqrt{N}}\sum_{i=1}^N \{A_1 \ldots A_i^* \ldots A_N 0_c\}\right] \quad (2)$$

and seen to be superpositions of electronic states with one cavity photon and electronic states without cavity photon and are labeled upper and lower polariton states. One of the dark states takes on the appearance

$$\Phi_d = \frac{1}{\sqrt{N}}\sum_{i=1}^N (-1)^i \{A_1 \ldots A_i^* \ldots A_N 0_c\}, \quad (3)$$

where, for simplicity of presentation, we have chosen $N$ to be even. Then, $\sum_{i=1}^N (-1)^i = 0$, and the other dark states can be obtained by permuting the $N/2$ minus signs such that the $N-1$ dark states are orthogonal to each other. The dark states do not contain configurations with cavity photons and the effect of the cavity is to create 'traceless' superpositions of the zero photon configurations. To better understand the notion of dark and bright, one notices that the transition matrix element with any one-atom operator $\hat{O} = \sum_{i=1}^N \hat{o}_i$ between the ground state $\Phi_0 = \{A_1 A_2 \ldots A_N 0_c\}$ of the ensemble-cavity system and a dark state vanishes: $\langle \Phi_0 | \hat{O} | \Phi_d \rangle = 0$. In contrast, the transition moment of a polariton state takes on the value $|\langle \Phi_0 | \hat{O} | \Phi_{up/lp}\rangle|^2 = N|\langle A|\hat{o}|A^*\rangle|^2/2$ as if all the available transition moments of all atoms are shared by the two polariton states. Since the coupling of the atoms to an external laser field is by a one-atom operator, the dark states are not populated by the laser while the polariton states are efficiently populated. The fact that the energies of the latter are separated from the former is favorable for populating the polariton states.

We now introduce a foreign atom $B$, which we name impurity into the cavity. Before discussing the ICD in the cavity, we first consider the known situation of a single atom $A$ and a neighbor $B$ in the absence of a cavity. For ICD to be operative the excitation energy $E_A$ of $A$ must exceed the ionization potential of $B$ ($IP_B$). Then, we have

$$A^* \cdots B \rightarrow A \cdots B^+ + e_{ICD}, \quad (4)$$

where $e_{ICD}$ stands for the electron emitted by ICD, briefly, the ICD electron. The kinetic energy of this electron is $E_A - IP_B$. At large interatomic distance $R$ between $A$ and $B$ and dipole allowed

transition $A^* \rightarrow A$, the ICD rate takes on the appearance $\Gamma = 2\pi|\gamma/R^3|^2$, where the decay amplitude $\gamma$ can be expressed as the sum of products of amplitudes describing processes on the individual atoms, i.e., the deexcitation of atom $A$ and the ionization of atom $B$[52,53]. For closed-shell atoms $A$ and $B$, one finds

$$\gamma = \mathbf{D}_A \cdot \mathbf{D}_B - 3(\mathbf{D}_A \cdot \mathbf{u})(\mathbf{D}_B \cdot \mathbf{u}), \tag{5}$$

where $\mathbf{u}$ is the unit vector pointing from $B$ to $A$, $\mathbf{D}_A = \langle A^*|\mathbf{d}_A|A\rangle$ is the dipole transition matrix element for the deexcitation of $A$ and $\mathbf{D}_B = \langle B|\mathbf{d}_B|B^+\rangle$ is that for the ionization of $B$. Here, $|B\rangle$ is the initial state of $B$ before ICD took place, $|B^+\rangle$ is the ion including the emitted electron produced by ICD, and $\mathbf{d}_{A(B)}$ are the dipole operators of $A(B)$.

It is relevant to note that the transition driven by the dipoles parallel to the interatomic axis ($\sigma$ transition) gives rise to a decay amplitude, which is twice as large as that driven by the dipoles perpendicular to this axis ($\pi$ transition), i.e., $\gamma_\sigma = -2\gamma_\pi$, and, consequently, the respective ICD rates fulfill $\Gamma_\sigma = 4\Gamma_\pi$[52]. This phenomenon is counterintuitive at first sight, as it holds at large separations where the atoms are expected to be independent of each other.

We return to the ensemble of atoms $A$ and the cavity. For that purpose, we set the atom $B$ at the origin of a coordinate system, choose the polarization direction $\epsilon_c$ of the cavity as the $Z$ axis, and assign an index $i$ to the unit vector $\mathbf{u}_i$ pointing from $B$ to the $i$-th atom $A_i$ of the ensemble. The situation is depicted in Fig. 1. Employing spherical coordinates, each unit Cartesian vector becomes as usual $\mathbf{u}_i = (\cos(\phi_i)\sin(\theta_i), \sin(\phi_i)\sin(\theta_i), \cos(\theta_i))$. Due to the cavity, the dipole transition elements $\mathbf{D}_{A_i}$ of all atoms $A$ point parallel to the $Z$ axis and having the same absolute value can be written as $\mathbf{D}_{A_i} = D_A(0, 0, 1)$. To proceed, we construct for each atom pair $A_i-B$ the decomposition of $\mathbf{D}_{A_i}$ into its components parallel and perpendicular to the respective unit vector:

$$\mathbf{D}_{A_i}^\| = D_A \cos(\theta_i)(\cos(\phi_i)\sin(\theta_i), \sin(\phi_i)\sin(\theta_i), \cos(\theta_i)),$$
$$\mathbf{D}_{A_i}^\perp = D_A \sin(\theta_i)(-\cos(\phi_i)\cos(\theta_i), -\sin(\phi_i)\cos(\theta_i), \sin(\theta_i)). \tag{6}$$

Now we are in the position to compute the ICD rate of the ensemble in the cavity. As done for a single pair of atoms, one

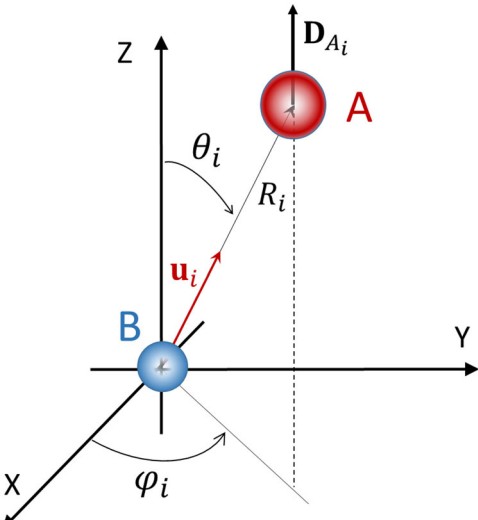

**Fig. 1 Coordinate system and vectors used.** The atom $B$ is at the origin and its distance to an atom $A_i$ of the ensemble is $R_i$. The unit vector $\mathbf{u}_i$ points from $B$ to $A_i$ and its spherical coordinates are $\theta_i$ and $\phi_i$. The transition dipole $\mathbf{D}_{A_i}$ of $A_i$ is parallel to the $Z$ axis and the polarization axis of the cavity.

starts from the golden rule[54]

$$\Gamma = 2\pi\sum_f |\langle\Psi_i|V|\Psi_f\rangle|^2, \tag{7}$$

where $V$ is the interaction between the atoms $A$ and the impurity $B$. The wavefunctions $\Psi_i$ and $\Psi_f$ describe as usual the initial and final states of the process in the absence of this interaction. For a single pair, the initial state is given by the product $\Psi_i = \{A^*\}\{B\}$ and the final state by $\Psi_f = \{A\}\{B^+\}$, and the golden rule has lead to the rate $\Gamma = 2\pi|\gamma/R^3|^2$ with the amplitude $\gamma$ presented in Eq. (5)[21,52]. For the ensemble in the cavity, the initial and final wavefunctions take on the appearance $\Psi_i = \Phi_{\mathrm{up/lp}}\{B\}$ and $\Psi_f = \Phi_0\{B^+\}$ and analogously for the dark states. In complete analogy to the pair of atoms, the golden rule leads to the following relations for the polariton states

$$\Gamma = \frac{2\pi}{2N}\left|\sum_{i=1}^N \gamma_i/R_i^3\right|^2, \tag{8}$$
$$\gamma_i = \mathbf{D}_{A_i}^\| \cdot \mathbf{D}_B - 3D_A\cos(\theta_i)\mathbf{D}_B \cdot \mathbf{u}_i + \mathbf{D}_{A_i}^\perp \cdot \mathbf{D}_B,$$

where $R_i$ is the interatomic distance between atom $A_i$ of the ensemble and $B$.

Since $\mathbf{D}_B$ is a vector, it is useful to first investigate the relevant quantity $\sum_{i=1}^N \gamma_i/R_i^3$ in the above equation separately for its three basis vectors in $X$, $Y$, and $Z$ directions, which we just call $S_X$, $S_Y$, and $S_Z$. Choosing $\mathbf{D}_B = D_B(0, 0, 1) = D_B\mathbf{e}_z$, and similarly for the other directions, leads with the aid of the explicit expressions in Eq. (6) to

$$S_Z = -\gamma_\pi \sum_{i=1}^N \left[\frac{3\cos^2(\theta_i) - 1}{R_i^3}\right],$$
$$S_X = -\frac{3\gamma_\pi}{2}\sum_{i=1}^N \left[\frac{\sin(2\theta_i)\cos(\phi_i)}{R_i^3}\right], \tag{9}$$
$$S_Y = -\frac{3\gamma_\pi}{2}\sum_{i=1}^N \left[\frac{\sin(2\theta_i)\sin(\phi_i)}{R_i^3}\right].$$

$Z$ is the polarization direction of the cavity and $S_Z$ is seen to depend only on the $\theta_i$ angles of the atoms of the ensemble. Notice that $\gamma_\pi$ contains all atom-specific quantities entering the expression for the decay amplitude of a single pair in the absence of the cavity.

The final transition matrix element for the ionization of $B$ and with it the decay rate can be written as

$$\mathbf{D}_B = D_B\left[S_X\mathbf{e}_x + S_Y\mathbf{e}_y + S_Z\mathbf{e}_z\right]/C,$$
$$\Gamma = \frac{2\pi}{2N}\left[|S_X|^2 + |S_Y|^2 + |S_Z|^2\right], \tag{10}$$

where $C = 1/\left[|S_X|^2 + |S_Y|^2 + |S_Z|^2\right]^{1/2}$ is a normalization constant.

From the above equations, it can already be anticipated that the ICD process in a cavity is highly sensitive to the geometrical arrangement of the atoms of the ensemble. We shall also see that the location of the impurity with respect to the atoms of the ensemble plays a crucial role in the ICD process and this well beyond the trivial fact that ICD depends on the distance between the atoms of the ensemble and the impurity. Let us start the discussion by putting the ensemble and the impurity in the plane perpendicular to the polarization direction of the cavity. In this simple scenario, all $\theta_i$ angles are $\pi/2$ and $S_X = S_Y = 0$ and the ICD rate is

$$\Gamma = \frac{\Gamma_\pi}{2N}\left|\sum_{i=1}^N (R/R_i)^3\right|^2, \tag{11}$$

where $\Gamma_\pi$ is the ICD rate of a single pair at distance $R$ in the absence of cavity. Clearly, the ICD rate of the ensemble in the

cavity depends only on the distribution of the distances between the $A$ atoms and the impurity. In general, the atoms which are close to the impurity contribute most to the rate, and because of the $2N$ denominator, the ICD rate is expected to be small for large numbers $N$ of atoms. That entanglement can stabilize the ensemble against ICD can also be nicely observed for an ordered ensemble. Consider a linear chain of $N$ atoms $A$ lying in the $XY$-plane with interatomic distance R between adjacent atoms and the central atom is replaced by the impurity $B$. It is easy to evaluate the rate with the above equation. For $N = 2$, i.e., one $A$ atom on each side of $B$, the rate is $\Gamma = \Gamma_\pi$ and thus as large as that without the cavity, but the situation changes as $N$ grows. For large enough $N$ (assuming two equal chains of $N/2$ atoms on both sides of $B$) one obtains $\Gamma = 2.88\,\Gamma_\pi/N$, which can be rather small for large $N$.

The situation changes dramatically when atoms $A$ form a ring around $B$. Then, all atoms $A_i$ have the same distance $R$ from the impurity and the ICD rate of a polariton state grows linearly with $N$:

$$\Gamma = \Gamma_\pi N/2. \tag{12}$$

Each of the polariton states shares half of the maximally possible decay rate where each $A$ atom contributes $\Gamma_\pi$ to the decay. As a consequence one notes that a dark state of the ring ensemble cannot decay by ICD and its decay rate vanishes. This also follows from the golden rule, Eq. (7), using $\Psi_i = \Phi_d\{B\}$.

We continue with the ring ensemble and shift the impurity vertically out of the center of the ring (Fig. 2b). Again, all atoms of the ensemble have the same angle $\theta$, but this angle depends on the distance of the impurity $B$ from the center of the ring. Consequently, in general $S_X$ and $S_Y$ do not vanish unless the atoms of the ensemble are equidistantly located on the ring. In this equidistant case, the rate follows, see Eqs. ((8) and (9)), a particularly simple expression

$$\Gamma = \frac{\Gamma_\pi N}{2}\left[3\cos^2(\theta) - 1\right]^2, \tag{13}$$

where $\Gamma_\pi$ is the rate of a single pair undergoing ICD without the cavity. It is seen that due the entanglement of the ensemble's atoms there is an explicit dependence of the rate on the second Legendre polynomial in $\theta$. This causes the rate to disappear at the magic angle $\theta = 54.74°$. Magic angles appear in many areas of physics like in photoionization[55] and NMR[56]. In the present context, the ring becomes stable against decay by ICD at the magic angle.

Next, we depart from the ensemble being confined to a plane perpendicular to the polarization direction of the cavity by tilting the ring with $B$ in its center. To be specific, the ring is rotated around the $Y$ axis by an angle $\alpha$ (Fig. 2c). Then, each point $(\cos(\phi_{pi}), \sin(\phi_{pi}), 0)$ on the planar ring becomes $(\cos(\alpha)\cos(\phi_{pi}), \sin(\phi_{pi}), \sin(\alpha)\cos(\phi_{pi}))$, which determines the $\theta_i, \phi_i$ angles needed to compute the ICD rates via Eqs. ((9) and (10)) of the tilted ring. This leads to

$$S_X = -\frac{3N\gamma_\pi}{4R^3}\sin^2(2\alpha), \qquad S_Y = 0,$$

$$S_Z = -\frac{N\gamma_\pi}{R^3}\left[3\sin^2(\alpha)/2 - 1\right], \tag{14}$$

$$\Gamma = \frac{N\Gamma_\pi}{2}\left[\frac{9}{16}\sin^4(2\alpha) + \left(\frac{3}{2}\sin^2(\alpha) - 1\right)^2\right].$$

In the derivation of the above equation, it has been assumed that the atoms of the ensemble are distributed equidistantly on the ring. If this is not the case, the expressions become more involved and show that the ICD decay reflects the distribution on the ring.

At zero tilt ($\alpha = 0$), the transition dipole $\mathbf{D}_B$ of the impurity points along the polarization axis of the quantized light, and the decay rate is $N\Gamma_\pi/2$. Tilting the ring now makes this dipole rotate

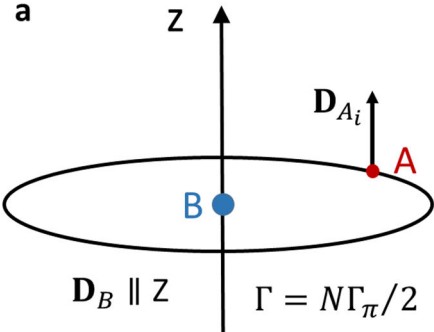

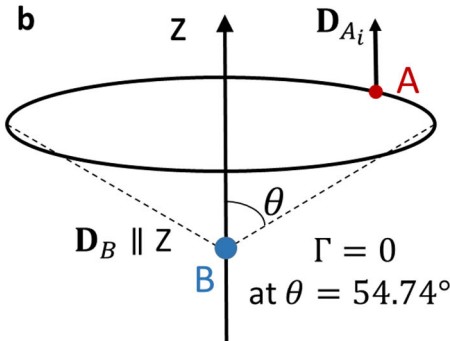

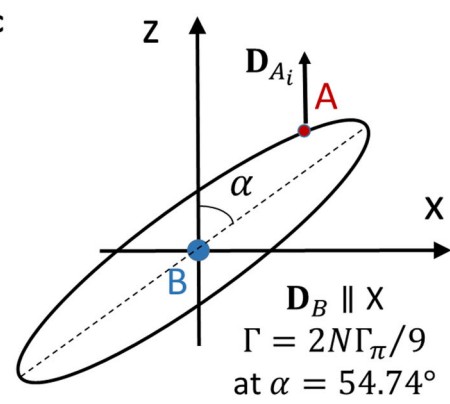

**Fig. 2 ICD in a ring-impurity ensemble. a** The ring is in the $XY$-plane with $B$ in the center. The ICD rate is strongly enhanced in the cavity from $\Gamma_\pi$ without a cavity to $N\Gamma_\pi/2$. **b** The ring is shifted along $Z$. The ICD rate now depends on the angle $\theta$ (Eq. 13). There is no ICD at all at the magic angle. **c** $B$ is in the center and the ring is rotated around the $Y$ axis. The transition dipole of $B$ rotates from the $Z$ axis into the $XZ$-plane and the ICD rate depends on the tilt angle $\alpha$ (Eq. 14). At the magic tilt angle, the transition dipole points parallel to the $X$ axis.

into the $XZ$-plane. Once the tilt arrives at the magic tilt angle ($\alpha = 54.74°$), the dipole in the polarization direction vanishes and now points completely in the $X$ direction and the rate becomes $2N\Gamma_\pi/9$. We mention here that not only the rate is a measurable quantity, but also the angular distribution of the emitted ICD electron is measurable, see, e.g., refs. [57,58], and this distribution depends on the direction of the transition dipole.

Atomic and molecular clusters have been subject to continuous interest over many years[59,60] and much interest has been devoted to their possible ground state geometrical structures and properties. Many ICD experiments have been carried out with rare gas clusters[21] and as the interaction of the atoms is weak this makes

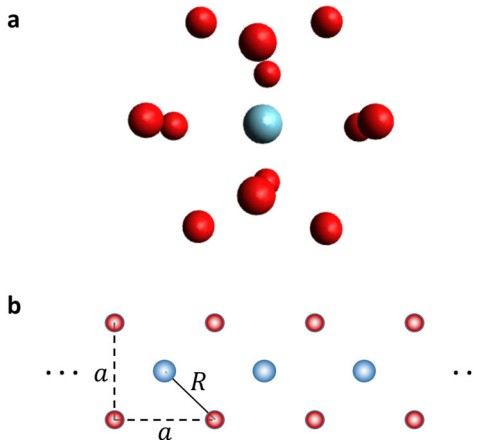

**a**

**b**

**Fig. 3 Cluster and 2D lattice structures considered. a** ArNe$_{12}$ cluster of icosahedral symmetry. The drawing is courtesy of *E. Fasshauer.* **b** A planar lattice of four *A* atoms surrounding each *B* atom. The lattice is perpendicular to the polarization direction of the cavity and the lattice constant is *a*.

them particularly suitable for cavity investigations. We consider here ArNe$_{12}$ as an example. Several energetically low-lying stable local structures are obtained by optimization with a universal force field[61]. The highest in symmetry is an icosahedral structure with a Ne–Ar distance $R = 3.406$ Å (Fig. 3a). Without a cavity, a weak external laser would excite a single Ne atom which undergoes ICD with the central Ar atom. For $2p \rightarrow 3s$ transition, the ICD rate $\Gamma_\pi$ corresponds to an ICD lifetime of 375 fs, which is nearly 4 orders of magnitude shorter than the radiative lifetime of 1.6 ns[53]. Putting the ArNe$_{12}$ cluster in the cavity with the highest symmetry axis along $Z$ (4 Ne atoms in each $XY$, $XZ$, and $YZ$ plane) and employing Eqs. (9) and (10) have surprised us considerably. The result is $\Gamma = 0$. That is, the cluster is stable against ICD.

We searched for the maximal rate by rotating the cluster (rotation around Z does not change the rate) and found that rotating around the $X$ axis by 45° gives rise to $\Gamma = 0.3\,\Gamma_\pi$ and the transition dipole $\mathbf{D}_B$ points parallel to the $Y$ axis. We mention here that as long as polaritonic states are formed, the above findings on the ICD rate are independent on the coupling strength $g$. However, available cavities do have sizeable losses and strong coupling is needed in order to form the polariton[62]. See also the discussion at the end. We also note that there are experimental investigations of ICD in much larger clusters, for instance, ICD in Ne clusters with about 5000 atoms[63], ICD in doped He nanodroplets with about 50,000 atoms[64], ICD in mixed NeKr clusters with about 1000 atoms[65] and ICD in He nanodroplets with about 10,000 and 50,000 atoms[66,67].

Finally, we address the issue of having more neighbors $B$. Since this is a whole subject by itself, we concentrate on one example, which shows how to compute the ICD rate and demonstrates a particular impact of the cavity on ICD. Consider the lattice with lattice constant $a$ in the plane perpendicular to the polarization direction shown in Fig. 3b, where each $B$ is surrounded by four equivalent $A$ atoms. To be able to employ the equations derived above for a single $B$, we assign the index $k$ to $B$ and note that one can compute all required quantities for each $B_k$ separately and obtain the partial rate $\Gamma_k$. For that purpose, $R_i$ becomes the distance $R_{ik}$ between $B_k$ and $A_i$, $\gamma_i$ becomes $\gamma_{ik}$, and so on. The total rate is then $\Gamma = \sum_{k=1}^{M} \Gamma_k$, where $M$ is the number of $B$ atoms. In the absence of cavity, a single atom $A$ is excited and its decay rate due to its two nearest neighbors at distance $a/\sqrt{2}$ is $2\Gamma_\pi$. The next pair of neighbors are at distance $a\sqrt{(5/2)}$ and contribute $2\Gamma_\pi/5^3$

to the rate and the next pair $2\Gamma_\pi/13^3$. Adding the whole series shows that compared to considering only the next neighbors, the total impact of all other neighbors leads to a minor enhancement of $8.5 \times 10^{-3}$%, which is rather negligible.

Now we consider the presence of the cavity and being in the $XY$-plane, we resort to Eq. (11) that gives the rate for a single $B_k$ and where $R_i$ is replaced by $R_{ik}$. To make contact with the cavity-free case, $R = a/\sqrt{2}$. Accordingly, $B_k$ has four nearest neighbors which, if the other $A$ atoms are not considered, gives rise to $\Gamma_k = \Gamma_\pi 4^2/(2N)$ and, for a long lattice, where $M = N/2$ and all $B$ atoms have the same rate, $\Gamma = 4\Gamma_\pi$, which is twice as much as without the cavity. What about including farther away neighbors in the cavity? Following Eq. (11) leads to the series $\Gamma_k = \Gamma_\pi/(2N)[4 + 4(1/5)^{3/2} + 4(1/13)^{3/2} + ...]^2$, which after multiplication with $N/2$ gives the total rate $\Gamma = 5\Gamma_\pi$. We see that the impact of more remote neighbors is much more important in cavity than in its absence.

## Discussion

There is a particularly important difference between the collective excitation in the cavity, i.e., polaritons, and a collective excitation of the atoms formed by a laser. The coupling strength to the cavity determines the energy of the polaritons and this can be used to control ICD. Imagine an excited atom $A^*$ whose excess energy (the energy difference between the excited and ground states) is somewhat smaller than the ionization energy of $B$ and thus insufficient to ionize an atom $B$. As a consequence, a collective excitation of $A$ atoms by a laser will not lead to ICD. This strongly contrasts with the potential a cavity has. Here, the energy of the upper polariton grows above the atomic excess energy as $\sqrt{N}g$ and may exceed the ionization energy of $B$ and ICD becomes operative. As two rare gas examples, we mention an ensemble of Ar atoms and Xe or Kr as a neighbor. For Ar$^*(3p \rightarrow 4s)$ the excess energy is 0.58 eV below the ionization energy of Xe, and for one of the Ar$^*(3p \rightarrow 3d)$ excited states it is just 0.02 eV below the ionization energy of Kr[68]. Another interesting scenario is met by choosing A$^*$ and B such that the ICD channel is open in the absence of cavity and the lower polariton is populated. Then, as the energy decreases as $-\sqrt{N}g$ below the atomic excess energy, one may by either increasing the number of atoms $A$ or/and increasing the coupling strength $g$ suddenly terminate the ICD process. Examples for this scenario could be an ensemble of Ne atoms and Ar as a neighbor like in the cluster discussed above, and an ensemble of Ar atoms and Kr as a neighbor. Here, the excess energy of Ne$^*(2s \rightarrow 3p)$ exceeds the ionization energy of Ar by 0.86 eV and that of Ar$^*(3p \rightarrow 5s)$ the ionization energy of Kr by just 0.07 eV[68]. The above-mentioned larger energy gaps of 0.58 and 0.86 eV cannot be overcome by the currently available cavity technology, see the discussion below. They are merely given to demonstrate the range of possibilities present even in simple, i.e., rare gas, atoms. Of course, other atoms and, in particular, molecules offer a larger variety for choosing candidates to open and close the ICD channel. Importantly, the cavity can also be used to manipulate the ICD rate in addition to switching on or off the ICD activity. The reason lies in the fact that the rate depends on the transferred energy and in the cavity this energy can be varied by changing the energy of the polariton. It is worth mentioning that the rate is typically largest at the ICD threshold where the transferred energy equals the ionization energy of the neighbor, see, e.g., refs. [5,6,21].

It is well known that in a laser field a coherent superposition state of $N$ atoms $A$ can be formed, which decay fast by spontaneous radiative emission, $N$ times faster than a single atom A (superradiance), see, e.g., ref. [69]. Since polaritons also include coherent superpositions (see, Eq. (2)), their spontaneous radiative

emission is also enhanced, but by a factor $N/2$. For an isolated $A-B$ dimer, the ICD is typically much faster than radiative decay[21]. In the cavity, as the distribution and orientation have been shown to play an important role, each situation deserves attention in order to know which channel is dominating. If we consider the $ArNe_{12}$ cluster as an example where ICD is particularly slow, we see (all numbers are given above) that the lifetime due to the radiative decay of the cluster in the cavity is 1.6 ns × $2/12 = 270$ ps. Although the ICD lifetime is just 375 fs/0.3 = 1.25 ps, it is still much shorter than the radiative lifetime.

Due to the entanglement of the atoms $A$ of an ensemble interacting with quantum light, the ICD process takes on very different features from classical ICD. One finds high sensitivity to the arrangement of the atoms and also to their orientation with respect to the polarization direction of the light, as well as to the position of the impurity $B$. Also, in cavity, the impact of more remote atoms and the related inclusion of farther neighbors can be substantially more important. Interestingly, symmetric arrangements like an endohedral icosahedrons can become ICD inactive. This calls for studies of clusters. Here, antisymmetric vibrational modes can cause ICD activity and the impact of interactions between the atoms has to be investigated. This all can make the field of clusters in cavity fruitful. It should be mentioned that not only the ICD rate is of relevance, but also the fact that ICD electrons are emitted and their energy and angular distribution are of relevance. As we have seen for the tilted ring, even the direction of the transition dipole varies strongly with the tilt angle.

Of course, one can expect similar effects for molecules. However, nuclear dynamics makes the molecule more complicated and as discussed in the introduction, molecules are more affected by the cavity since, e.g., light-induced conical intersections are created and such modifications must be included into the description of their dynamics. It has been recently shown that vibrational ICD[70] is efficient, where the excess vibrational energy of a molecule can be utilized to ionize a neighbor (e.g., anion). This reduces the involved energy substantially and enlarges the scope of ICD in the cavity. The present study makes clear that one can expect the related severe impact of quantum light also on other processes, which follow transition–dipole transition–dipole interactions. Here, we mention Foerster resonance energy transfer[71–73], and resonant[74,75] and non-resonant[76] vibrational energy transfer.

As we mentioned in the introduction, we concentrate in this first study on the fundamental aspect of the impact of quantum light on ICD and do not discuss details concerning the structure of the cavity. To take the structure of the cavity into account, a QED approach analogous to that in ref. [77] seems to be promising. Recently, this methodology was also used to investigate superradiant effects in resonant energy transfer in donor–acceptor ensembles, and interesting dependence on geometry was found[78]. Such an approach also includes the effect of retardation not included in the present study. Retardation makes the impact of ICD more long range. However, for the transfer energies discussed in this work the impact of retardation is very small.

Let us in the end touch upon the possibility to experimentally study the phenomena discussed in the present work. For this purpose, one will need set-ups in which strong-coupling regime with a small number of emitters can be achieved and kept long enough, that is, for times comparable with the ICD lifetime of the studied complex. Over the last decade, we have witnessed a tremendous progress of quantum cavity technologies and various resonators have already been developed, ranging from dielectric cavities[79] and surface evanescent modes[80] to plasmonic cavities with "sub-wavelength" mode volumes[81] (for a recent review, see, e.g., ref. [62]). Strong-coupling regimes enabling creation of polaritonic states even with a single molecule at room

temperature have already been reported[82]. Currently, these have been achieved using hybrid metallo-dielectric set-ups, in which a strongly sub-wavelength cavity can be formed through localized surface plasmon resonances in nanometer gaps. The lifetimes of these resonances are, however, typically on the order of 10 fs, which puts some restrictions on the ICD or other energy-transfer processes that can be studied. There are ICD processes faster than 10 fs[21], but we note that this does not mean that the only ICD processes with lifetimes below 10 fs can be addressed. The effect of the cavity on the ICD process will be measurable even if the ICD lifetime is longer, because we can compare the ICD yield in free space and in the cavity. We note also that, as discussed above, through the energy split between the upper and lower polaritons one may open or close an ICD process. We mention that energy splits of about 0.4 eV have been reported[82] even in the single-molecule regime. Moreover, both a decrease of the ICD lifetime and an increase of the light-matter coupling can be achieved by designing appropriate 2D structures, similar to those shown in Fig. 3b, in particular when molecules are involved.

The fast development of quantum-resonator technologies may substantially enlarge the range of experimentally accessible ICD processes and systems in the near future. We hope that the present study will further motivate the research in this direction.

## Data availability
Data sharing not applicable to this article as no datasets were generated or analyzed during the current study.

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

## Acknowledgements

We thank E. Fasshauer and K. Gokhberg for valuable contributions. Financial support by the European Research Council (ERC) (Advanced Investigator Grant No. 692657) is gratefully acknowledged.

## Author contributions

Both authors conceived the idea. L.S.C. derived the working equations. A.I.K. performed the calculations. Both authors analyzed the results and wrote the paper.

## Funding

## Competing interests

The authors declare no competing interests.
