## [Peer Review File · Nature Communications]

Reviewers' Comments:

Reviewer #1:

Remarks to the Author:

The authors consider the impact of an electromagnetic cavity on the process of interatomic Coulomb decay. This is certainly a timely topic due to the current interest on cavity-modified properties and processes. However, the manuscript has several fundamental problems that, in my opinion, make it unsuitable for publication in Nature Communications in its current form:

-) Most importantly, due to the approximations employed, the effect of the "cavity" is essentially reduced to that of having a collective initial excitation of the A atoms. However, that same excitation is also obtained in the same collection of atoms in free space by a laser pulse that acts on all atoms with the same phase and amplitude. Since the distances between the atoms considered here (while not given explicitly) are necessarily sub-wavelength (since otherwise, retardation effects in the atom-atom interaction should be taken into account), this is in fact simply the "natural" result of the dipole approximation, i.e., assuming that $\exp(ikr) \sim 1$ for all atoms. However, even for larger collections of atoms lying in a single plane, this would be obtained for plane-wave excitation under normal incidence to the plane. So the results that are presented, while interesting, seem to be simply results of treating ICD in collections of atoms and taking into account coherent excitation of the atoms, but have little to do with a cavity.

-) There is a large disconnect between the level of detail that is given for the abstract "cavity" mode and the details of atomic distribution etc. While one certainly cannot expect all parts of the system to be treated with the same level of detail, the approximations performed should at least be motivated and reasonable, particularly in light of the previous comment. Specifically, it is not discussed at all what cavity geometry the authors have in mind. The experimentally possible setups range from "normal" optical cavities (Fabry-Perot or similar) to plasmonic nanogap cavities, with significantly different physical properties and implications on the validity of different approximations. For example, optical cavities are necessarily "large" ($\lambda/2$) and only reach the strong coupling regime when filled with a large number of atoms (so the factor $1/N$ would make the rates obtained here vanishingly small), while plasmonic cavities are necessarily (highly) lossy and so small that the static Coulomb interaction will also be modified by the presence of metallic structure within a few nm. In general, depending on the properties of the cavity, one could imagine that long-range interactions due to propagation of (confined) photons could also become more important, such that the interaction decays much slower than the $1/R^3$ of the electrostatic term considered here.

-) Related to the above, for the ArNe₁₂ cluster that is discussed as an example, reaching strong coupling with this small number of atoms would require extremely small "cavities", with essentially the only available option being nanoplasmonic systems where the "cavity" modes are sub-wavelength localized surface plasmon resonances. This would also strongly modify electrostatic interactions.

-) For a consistent treatment of the effect of an arbitrary cavity structure, it seems like an approach based on macroscopic QED or similar (as, e.g., in Hemmerich et al., Nature Communications 9, 2934 (2018)), would be more appropriate. This could "automatically" include the effects of the chosen cavity structure on both propagating and electrostatic EM field.

Reviewer #2:

Remarks to the Author:

The manuscript by Cederbaum and Kuleff entitled "Impact of cavity on interatomic Coulombic decay" presents high novelty and significance in the field and I suggest its publication on Nature Communications with minor revision. The authors derive how the entanglement of the atoms of an ensemble interacting with quantum light makes the ICD process sensible to the orientation of the

atoms with respect to the polarization of the light, as well as to the distance with the impurity B and with respect to each others.

It would be interesting to know if the authors have tried other symmetries which, lead to the result $\Gamma=0$ like the icosahedral one resulting the system stable against ICD.

Below I have reported my suggestions aimed to improve the text for a sliding reading.

Page 3 line 6: Please add also the definition for the creation and annihilation operators a .

It would be recommended if the approximations leading to equation 1 should be explained in the supporting information starting from the more general definition of a total Hamiltonian in a cavity system. Alternatively the authors could indicate more references in which the derivation of equation 1 has already been treated.

Page 6 line 1: Please, put a cross reference after the words "golden rule" for a more clarity.

Page 6 line 22: the authors said: "Notice that γ_n is the decay amplitude of a single pair in the absence of the cavity". Please put a reference or explain it better. Choosing $DB=DB(0,0,1)$ it was not immediately clear to me to consider S_z depends on the decay amplitude driven by the dipole perpendicular to the interatomic axis γ_n

In FIG 2c insert the value of Γ as for the others.

Reference 9: Please correct reference 9 by putting the correct page range 508-511 and not 508511.

Page 7 line 14: please put a reference after the sentence "where Γ_n is the ICD rate of a single pair at distance R in the absence of cavity" or in alternative explain better the definition of cavity, Γ_n and R . I personally found difficulties by reading the second part of page 7 where, after equation 11, the authors make the example for $N=2$ in a linear configuration of A-B-A atoms separated by a distance R that was previously defined as the distance in the absence of cavity. Please can you explain me how did you get the value of $2\Gamma_n$? Considering B at the center of the axis and the two atoms A separated by R , I found $\Gamma = \Gamma_n$. I could think of it as a degenerate case of the ring where N is reduced to 2. Also the value of $\Gamma = 2.88 \Gamma_n/N$ is not clear to me. How does it come from? Maybe this part could be rephrased better. On the other side, the treatment for the ring is clear to me.

After equation 14 the authors declare: " In the derivation of the above equation it has been assumed that the atoms of the ensemble are distributed equidistantly on the ring". Having considered N to be even, I suppose that the authors want to specify that pairs of atoms A are symmetrically opposite to B in the plane. But I am not sure about it. Please can you explain me better the approximation?

Reviewer #3:

Remarks to the Author:

This theoretical manuscript considers the combination of two topical subjects: interatomic coulombic decay (ICD) - an ultrafast energy transfer process between two adjacent atoms A and B, which has been studied thoroughly in recent years and shown to be relevant in various areas of the natural sciences - and electromagnetic cavities, in which the quantum character of radiation modes plays a crucial role. The authors argue that the properties of ICD are strongly altered in the presence of a resonant single-photon mode and discuss the geometry dependence of ICD in an ensemble of $N+1$ atoms in a cavity.

From this point of view, the present study is of potential interest for Nat. Commun. The manuscript is well written, the applied methods are sound, and the derivations are presented in a transparent manner. However, before a final decision can be made, the authors are asked to consider the following remarks:

- The authors find that the rate for ICD in a cavity depends strongly on the number of atoms A and the geometry of their positions with respect to atom B. A similar collective behaviour is well known from spontaneous radiative decay of excited states in an ensemble of atoms. In particular, the phenomena of superradiance and subradiance may occur [e.g. Gross and Haroche, Phys. Rep. 93, 301 (1982)]. The authors of the present manuscript should add a discussion of this issue, especially because the spontaneous radiative decay competes with ICD. Therefore it matters which of the two decay channels is enhanced or suppressed more strongly under certain conditions.

- In a recent study, superradiant effects in resonant energy transfer between atoms, including the geometry dependence, have been investigated [arXiv:1912.05892]. In particular, the possibility for a complete suppression of the transfer rate inside a spherically symmetric distribution of donor atoms was found. The relation with this work needs to be discussed in the present manuscript.

- I wonder to which extent the predictions of the authors rely on the quantum description of the radiation field. It seems to me that similar - perhaps even identical - results would be obtained if the system of atoms was exposed to the coherent field of a classical electromagnetic (laser) wave, whose frequency is resonant to the atomic transition. Since this is a main point of the manuscript, I ask the authors to clearly indicate where their predictions based on quantum light differ from a semiclassical treatment.

- In line with my previous remark, I do not quite agree with the statement on page 10 that 'Without a cavity, a weak external laser would excite a single Ne atom'. In this situation the ensemble of N atoms A (after photoabsorption) would rather be in a coherent superposition state, as well. For the case $N=2$, corresponding expressions are given in the paper by Najjari, Muller and Voitkiv, New J. Phys. 14 (2012) 105028, which seems to be a follow-up on Ref.[44] with a detailed account of resonant photoionization in correlated three-atomic systems.

- Minor point: The summation index in eqs.(2) and (3) should be i instead of n . The same holds for some subsequent sums in the following text.

Replies to Reviewers' comments

Reviewer #1 (Remarks to the Author):

The authors consider the impact of an electromagnetic cavity on the process of interatomic Coulomb decay. This is certainly a timely topic due to the current interest on cavity-modified properties and processes. However, the manuscript has several fundamental problems that, in my opinion, make it unsuitable for publication in Nature Communications in its current form:

-) Most importantly, due to the approximations employed, the effect of the "cavity" is essentially reduced to that of having a collective initial excitation of the A atoms. However, that same excitation is also obtained in the same collection of atoms in free space by a laser pulse that acts on all atoms with the same phase and amplitude. Since the distances between the atoms considered here (while not given explicitly) are necessarily sub-wavelength (since otherwise, retardation effects in the atom-atom interaction should be taken into account), this is in fact simply the "natural" result of the dipole approximation, i.e., assuming that $\exp(ikr) \sim 1$ for all atoms. However, even for larger collections of atoms lying in a single plane, this would be obtained for plane-wave excitation under normal incidence to the plane. So the results that are presented, while interesting, seem to be simply results of treating ICD in collections of atoms and taking into account coherent excitation of the atoms, but have little to do with a cavity.

We agree that, in principle, a collective initial excitation of the A atoms can be obtained in free space by a laser pulse that acts on all atoms with the same phase and amplitude. This excitation is related, but not the "same excitation" as in a cavity, as indicated by the Referee. As seen, e.g., in Eq. (2), the collective excitation in the cavity is only a part of the polariton's wavefunction in the cavity. Similarly to superradiant states, the transition moment of a polariton state takes on a value much larger than that of an atom as if all the available transition moments of all atoms is shared by the two polariton states. However, the other states formed by the cavity are dark states and formally inaccessible directly, while many of the superposition states of free atoms formed by a laser other than the superradiant state are not dark. What is particularly important in the cavity is that the energy of the polariton states is well separated from the atomic energies and can thus be addressed directly by the laser exciting it. Consequently, a polariton state is much easier to selectively populate than a collective excitation of all atoms in a laser pulse.

Polaritonic states are formed in cavity and it is legitimate, relevant and timely to investigate ICD in these states, independently of whether other kinds of collective excitations can be formed in other contexts. Even more so, as transition dipole – transition dipole interactions are not only relevant for ICD but for other scenarios as well. We agree that we have missed to discuss the relationship to the collective excitation by a laser, which we do now.

Apart from the accessibility of polaritons discussed above there is a particularly important difference between collective excitation in cavity, i.e., polaritons, and collective excitation by a laser. The coupling strength to the cavity determines the energy of the polaritons and this can be used to control ICD. Imagine an excited atom A^* whose excess energy (the energy difference between the excited and

ground states) is somewhat smaller than the ionization energy of B and thus insufficient to ionize an atom B. As a consequence, a collective excitation of A atoms by a laser will not lead to ICD. This strongly contrasts the potential a cavity has. Here, the energy of the upper polariton grows above the atomic excess energy as $N^{1/2}g$ and may exceed the ionization energy of B and ICD becomes operative. Another interesting scenario is met by choosing A^* and B such that the ICD channel is open in the absence of cavity and the lower polariton is populated. Then, as the energy decreases as $-N^{1/2}g$ below the atomic excess energy, one may by either increasing the number of atoms A or/and increasing the coupling strength g suddenly terminate the ICD process. Importantly, the cavity can also be used to manipulate the ICD rate in addition to switching on or off the ICD activity. The reason lies in the fact that the rate depends on the transferred energy and in cavity this energy can be varied by changing the energy of the polariton. It is worth mentioning that the rate is typically largest at the ICD threshold where the transferred energy equals the ionization energy of the neighbor.

We thank the Reviewer for his/her comment which has led us to stress more clearly the usefulness of controlling ICD by having a cavity.

Several changes have been made to the manuscript.

At the end of the Abstract, we have added the sentence: "It is stressed that in contrast to superposition states formed by a laser, forming polaritons by a cavity enables to control the emergence and suppression as well as the efficiency of ICD."

Above equation (1) the following sentence has been added: "Since the energies of polaritonic states can be manipulated, we shall see that cavities enable opening and closing the ICD channel which is not possible for a superposition state formed by a laser."

A new paragraph has been inserted and is now the first new paragraph on page 12: "There is a particularly important difference between the collective excitation in cavity, i.e., polaritons, and a collective excitation of the atoms formed by a laser. The coupling strength to the cavity determines the energy of the polaritons and this can be used to control ICD. Imagine an excited atom A^* whose excess energy (the energy difference between the excited and ground states) is somewhat smaller than the ionization energy of B and thus insufficient to ionize an atom B. As a consequence, a collective excitation of A atoms by a laser will not lead to ICD. This strongly contrasts the potential a cavity has. Here, the energy of the upper polariton grows above the atomic excess energy as $N^{1/2}g$ and may exceed the ionization energy of B and ICD becomes operative. As two rare gas examples we mention an ensemble of Ar atoms and Xe or Kr as a neighbor. For $Ar^*(3p \rightarrow 4s)$ the excess energy is 0.58 eV below the ionization energy of Xe, and for one of the $Ar^*(3p \rightarrow 3d)$ excited states it is just 0.02 eV below the ionization energy of Kr [62]. Another interesting scenario is met by choosing A^* and B such that the ICD channel is open in the absence of cavity and the lower polariton is populated. Then, as the energy decreases as $-N^{1/2}g$ below the atomic excess energy, one may by either increasing the number of atoms A or/and increasing the coupling strength g suddenly terminate the ICD process. Examples for this scenario could be an ensemble of Ne atoms and Ar as a neighbor like in the cluster discussed above, and an ensemble of Ar atoms and Kr as a neighbor. Here, the excess energy of $Ne^*(2s \rightarrow 3p)$ exceeds the ionization energy of Ar by 0.86 eV and that of $Ar^*(3p \rightarrow 5s)$ the ionization energy of Kr by just 0.07 eV [62]. Of course, other atoms and, in particular, molecules offer a larger variety for choosing candidates to open and close the ICD channel. Importantly, the cavity can also be used to manipulate the ICD rate in addition to switching on or off the ICD activity. The reason lies in the fact that the rate depends on the

transferred energy and in cavity this energy can be varied by changing the energy of the polariton. It is worth mentioning that the rate is typically largest at the ICD threshold where the transferred energy equals the ionization energy of the neighbor, see, e.g., [5,21,63]. ”

-) There is a large disconnect between the level of detail that is given for the abstract "cavity" mode and the details of atomic distribution etc. While one certainly cannot expect all parts of the system to be treated with the same level of detail, the approximations performed should at least be motivated and reasonable, particularly in light of the previous comment. Specifically, it is not discussed at all what cavity geometry the authors have in mind. The experimentally possible setups range from "normal" optical cavities (Fabry-Perot or similar) to plasmonic nanogap cavities, with significantly different physical properties and implications on the validity of different approximations. For example, optical cavities are necessarily "large" ($\lambda/2$) and only reach the strong coupling regime when filled with a large number of atoms (so the factor $1/N$ would make the rates obtained here vanishingly small), while plasmonic cavities are necessarily (highly) lossy and so small that the static Coulomb interaction will also be modified by the presence of metallic structure within a few nm. In general, depending on the properties of the cavity, one could imagine that long-range interactions due to propagation of (confined) photons could also become more important, such that the interaction decays much slower than the $1/R^3$ of the electrostatic term considered here.

We are aware of the fact that there are different kinds of cavities available. Since it is the first work on the subject, we concentrate here on the fundamental aspect of the impact on ICD and do not discuss technical details concerning the choice of the cavity. We would like to mention that many recent works in the literature (e.g., PRL, ref. [42], PNAS, ref. [25], PRX, ref. [47]) study fundamental aspects of systems in cavity without taking into account the details of the cavity. In future work on explicit applications, aspects of the cavity used will have to be included.

The issue of the slower decay than that of the electrostatic term addressed by the Reviewer is discussed in the last point below in connection with the QED approach.

We inserted a sentence into the paragraph above Eq. (1): “There are several types of cavities available nowadays, but in this first study we concentrate on the fundamental aspect of the impact of quantum light on ICD and do not discuss details concerning the choice of the cavity.”

-) Related to the above, for the ArNe₁₂ cluster that is discussed as an example, reaching strong coupling with this small number of atoms would require extremely small "cavities", with essentially the only available option being nanoplasmonic systems where the "cavity" modes are sub-wavelength localized surface plasmon resonances. This would also strongly modify electrostatic interactions.

As long as polariton states are formed and they are addressable, there is no need for reaching strong coupling, the findings on the ICD rate are independent on the coupling strength g . In addition, there are experimental investigations of ICD on much larger clusters, for instance, ICD in Ne clusters with about 5000 atoms [K. Nagaya et al., Nature Commun. **7**, 13477 (2016)], ICD in doped He nanodroplets with about 50000 atoms [A. C. LaForge et al., Nat. Phys. **15**, 247 (2019)], ICD in mixed NeKr clusters with about 1000 atoms [T. Arion et al., JCP **134**, 074306 (2011)] and ICD in He nanodroplets with about 50000

atoms [Y. Ovcharenko et al., PRL **121**, 073401 (2014)]. Such clusters can also serve as examples in cavity. Even if one uses small cavities, the electrostatic interactions can be expected to still provide the leading term and one will see differences according to the arrangement and orientation of the cluster, what is an important outcome of this work.

At the end of the discussion of the example of ArNe₁₂ we added the text: “We mention here that as long as polaritonic states are formed and are addressable, there is no need for reaching strong coupling, as the above findings on the ICD rate are independent on the coupling strength g .”

-) For a consistent treatment of the effect of an arbitrary cavity structure, it seems like an approach based on macroscopic QED or similar (as, e.g., in Hemmerich et al., Nature Communications 9, 2934 (2018)), would be more appropriate. This could “automatically” include the effects of the chosen cavity structure on both propagating and electrostatic EM field.

The derivation by Hemmerich *et al.* is well known to us and we appreciate it. They also start from the golden rule and succeeded to express the ICD rate for the transition matrix element of higher order by a dyadic Green’s tensor. It is rather complicated to compute this tensor for a given cavity structure. In vacuum they obtain for the rate $\Gamma(R) = \Gamma^{\text{NR}}(R) [1 + x^2/3 + x^4/3]$ where $x = R\omega/c$ ($\hbar\omega$ is the transferred energy in ICD and c speed of light) and $\Gamma^{\text{NR}}(R)$ is the ICD rate without retardation. The retardation indeed enhances the decay at larger distances R as seen in the quotient $\Gamma(R)/\Gamma^{\text{NR}}(R) = 1 + x^2/3 + x^4/3$. However, the distances where the impact of retardation is at all noticeable are so large at the energies addressed in this work that the total rate $\Gamma(R)$ is itself already vanishingly small as $\Gamma^{\text{NR}}(R)$ is proportional to $1/R^6$ at large R . Let us give an example of an excess energy $\hbar\omega$ of 10 eV and a distance R of 1 nm. Then, the enhancement due to retardation is 8.5×10^{-4} , i.e., negligible. For the retardation to take on the small value of just 1% enhancement, one has to go to the distance of 3.4 nm at which the ICD rate has dropped by a factor of 6.5×10^{-4} from its value at 1 nm and became rather irrelevant.

The non-retarded rate is the leading term for the issues addressed in our work and we would like to keep a simpler line of derivation for the sake of readability. In future work it would be interesting to address and take into account a specific cavity, but here we would like to concentrate on the fundamental aspect of what impact is to be expected for ICD in a polaritonic state.

At the end of the manuscript we added the following discussion (see also reply to Reviewer #3): “As we mentioned in the introduction, we concentrate in this first study on the fundamental aspect of the impact of quantum light on ICD and do not discuss details concerning the choice of the cavity. To take the structure of the cavity into account, a QED approach analogous to that in Ref. [72] seems to be promising. Recently, this methodology was also used to investigate superradiant effects in resonant energy transfer in donor-acceptor ensembles and interesting dependence on geometry was found [73]. Such an approach also includes the effect of retardation not included in the present study. Retardation makes the impact of ICD more long range. However, for the transfer energies discussed in this work the impact of retardation is very small.”

Reviewer #2 (Remarks to the Author):

The manuscript by Cederbaum and Kuleff entitled "Impact of cavity on interatomic Coulombic decay" presents high novelty and significance in the field and I suggest its publication on Nature Communications with minor revision. The authors derive how the entanglement of the atoms of an ensemble interacting with quantum light makes the ICD process sensible to the orientation of the atoms with respect to the polarization of the light, as well as to the distance with the impurity B and with respect to each others.

We thank the Referee for highly praising our work.

It would be interesting to know if the authors have tried other symmetries which, lead to the result $\Gamma = 0$ like the icosahedral one resulting the system stable against ICD.

As we show in our work, there are different arrangements of collectively excited atoms in a cavity which are stabilized against ICD with an impurity due to destructive interference of the individual decay amplitudes. Such an arrangement is, for example, the ring of excited atoms shifted along the polarization direction such that it makes/forms a magic angle with the impurity atom. Within this first study we did not encounter other examples, but it is clear that with the help of the derived in our work explicate expressions for the decay rate and its dependence on the mutual orientation of the polarization direction of the cavity and the transition dipoles of the initially excited ensemble and the impurity, one will be able to design other ICD inactive arrangements.

Below I have reported my suggestions aimed to improve the text for a sliding reading.

Page 3 line 6: Please add also the definition for the creation and annihilation operators a . It would be recommended if the approximations leading to equation 1 should be explained in the supporting information starting from the more general definition of a total Hamiltonian in a cavity system. Alternatively the authors could indicate more references in which the derivation of equation 1 has already been treated.

The corresponding operators are the bosonic creation and annihilation operators of the cavity field mode. The Hamiltonian given in Eq. (1) is quite standard and appears in many papers and books, some of which we cite ("[43,46-48]"). Its derivation can be found, for example, in the book of Faisal, Ref. [46]. We agree, however, with the Referee that all quantities appearing in the Hamiltonian need to be clearly defined and explained. That is why, we modified the corresponding text as follows:

"... where $H_e = \sum_{i=1}^N H_i$ is the electronic Hamiltonian of the ensemble, $\vec{d} = \sum_{i=1}^N \vec{d}_i$ is the total dipole operator of the ensemble, g is the coupling strength between the cavity and the atoms, and \hat{a}^\dagger and \hat{a} are creation and annihilation operators for the bosonic electric field mode."

Page 6 line 1: Please, put a cross reference after the words “golden rule” for a more clarity.

We now refer to the review of Bambynek *et al.*, Ref. [54] in the revise manuscript, where the Fermi golden rule is discussed also in the context of electronic decay.

New reference: W. Bambynek, B. Crasemann, R. W. Fink, H.-U. Freund, H. Mark, C. D. Swift, R. E. Price, and P. V. Rao, *Rev. Mod. Phys.* **44**, 716 (1972).

Page 6 line 22: the authors said: “Notice that $\Upsilon\pi$ is the decay amplitude of a single pair in the absence of the cavity”. Please put a reference or explain it better. Choosing $DB=DB(0,0,1)$ it was not immediately clear to me to consider S_z depends on the decay amplitude driven by the dipole perpendicular to the interatomic axis $\Upsilon\pi$

To make it clearer we reformulated the corresponding text as follows:

“...Notice that γ_π contains all atom-specific quantities entering the expression for the decay amplitude of a single pair in the absence of the cavity.”

In FIG 2c insert the value of Γ as for the others.

We now give the explicit expression for Γ also in Fig. 2c.

Reference 9: Please correct reference 9 by putting the correct page range 508-511 and not 508511.

We thank the Referee for pointing out this inconsistency in the bibtex compilation. All typos like that in the reference list are now corrected.

Page 7 line 14: please put a reference after the sentence “where $\Gamma\pi$ is the ICD rate of a single pair at distance R in the absence of cavity” or in alternative explain better the definition of cavity, $\Gamma\pi$ and R . I personally found difficulties by reading the second part of page 7 where, after equation 11, the authors make the example for $N=2$ in a linear configuration of A-B-A atoms separated by a distance R that was previously defined as the distance in the absence of cavity. Please can you explain me how did you get the value of $2\Gamma\pi$? Considering B at the center of the axis and the two atoms A separated by R , I found $\Gamma = \Gamma\pi$. I could think of it as a degenerate case of the ring where N is reduced to 2. Also the value of $\Gamma = 2.88 \Gamma\pi/N$ is not clear to me. How does it come from? Maybe this part could be rephrased better. On the other side, the treatment for the ring is clear to me.

The Referee is correct and indeed in A–B–A system there is no change in the rate compared to the ICD in a single A–B pair without a cavity, i.e. $\Gamma = \Gamma_\pi$. We thank the Referee for pointing out this mistake. The expression for the large N case is, however, correct and obtained from Eq. (11) as follows. Assuming two equal chains of $N/2$ atoms on both sides of B, we have $\Gamma = \frac{\Gamma_\pi}{2N} \left| 2 \sum_{i=1}^{N/2} \frac{1}{i^3} \right|^2$. The i^{-3} series converges to the Apéry’s constant, having an approximate value of 1.20206.

To reflect those issues, we made the following changes to the corresponding text in the manuscript:

“For $N = 2$, i.e., one A atom on each side of B, the rate is $\Gamma = \Gamma_{\pi}$ and thus as large as that without the cavity, but the situation changes as N grows. For large enough N (assuming two equal chains of $N/2$ atoms on both sides of B) one obtains $\Gamma = 2.88 \Gamma_{\pi}/N$ which can be rather small for large N .”

After equation 14 the authors declare: “ In the derivation of the above equation it has been assumed that the atoms of the ensemble are distributed equidistantly on the ring”. Having considered N to be even, I suppose that the authors want to specify that pairs of atoms A are symmetrically opposite to B in the plane. But I am not sure about it. Please can you explain me better the approximation?

We mean that the distance between every two adjacent atoms A on the ring is the same. This indeed implies that in the case of even N , every atom A has a counterpart on the opposite side of the ring. Such an arrangement helps to take the sum/integral over θ and ϕ , and thus to arrive at the closed expressions given in Eq. (14). Other (e.g. inhomogeneous) distributions of the particles on the ring will lead to more involved expressions, which will reflect the particular distribution and geometry.

Reviewer #3 (Remarks to the Author):

This theoretical manuscript considers the combination of two topical subjects: interatomic coulombic decay (ICD) - an ultrafast energy transfer process between two adjacent atoms A and B, which has been studied thoroughly in recent years and shown to be relevant in various areas of the natural sciences - and electromagnetic cavities, in which the quantum character of radiation modes plays a crucial role. The authors argue that the properties of ICD are strongly altered in the presence of a resonant single-photon mode and discuss the geometry dependence of ICD in an ensemble of N+1 atoms in a cavity.

From this point of view, the present study is of potential interest for Nat. Commun. The manuscript is well written, the applied methods are sound, and the derivations are presented in a transparent manner. However, before a final decision can be made, the authors are asked to consider the following remarks:

- The authors find that the rate for ICD in a cavity depends strongly on the number of atoms A and the geometry of their positions with respect to atom B. A similar collective behaviour is well known from spontaneous radiative decay of excited states in an ensemble of atoms. In particular, the phenomena of superradiance and subradiance may occur [e.g. Gross and Haroche, Phys. Rep. 93, 301 (1982)]. The authors of the present manuscript should add a discussion of this issue, especially because the spontaneous radiative decay competes with ICD. Therefore it matters which of the two decay channels is enhanced or suppressed more strongly under certain conditions.

In a coherent superposition state of N atoms A formed by a laser the decay rate by spontaneous radiative emission is N times larger than that of a single atom A and the resulting short radiative lifetime may indeed compete with the ICD, in particular, when the ICD rate of the coherent superposition of the ensemble is small like in the ArNe₁₂ cluster or ring tilted by an angle close to the magic angle. In cavity the decay rate by spontaneous radiative emission is N/2 times larger than that of a single atom A as follows from the wavefunction of a polariton (see Eq. (2) in the manuscript). The factor N/2 also enters, however, the ICD rate, see, e.g., Eq. (12). If we consider the ArNe₁₂ cluster as an example where ICD is particularly slow, we see (all numbers are given the manuscript) that the lifetime due to the radiative decay of the cluster in cavity is $1.6 \text{ ns} \times 2/12 = 0.27 \text{ ns}$. Although the ICD lifetime is just $375 \text{ fs} / 0.3 = 1.25 \text{ ps}$, it is still much shorter than the radiative lifetime.

To clarify the issue, we added the following text into the revised manuscript.

Above equation (1) the following sentence has been added: "There is resemblance between polaritonic states and coherent superposition states of an ensemble formed by a laser which will be addressed after we have introduced and applied the ICD to polaritonic states."

After the ICD examples, the following paragraph has been added: "It is well known that in a laser field a coherent superposition state of N atoms A can be formed which decay fast by spontaneous radiative emission, N times faster than a single atom A (superradiance), see, e.g., [64]. Since polaritons also include coherent superpositions (see, Eq. (2)), their spontaneous radiative emission is also enhanced, but by a factor N/2. For an isolated A-B dimer, the ICD is typically much faster than radiative decay [21]. In cavity, as the distribution and orientation has been shown to play an important role, each situation deserves attention in order to know which channel is dominating. If we consider the ArNe₁₂ cluster as an

example where ICD is particularly slow, we see (all numbers are given above) that the lifetime due to the radiative decay of the cluster in cavity is $1.6 \text{ ns} \times 2/12 = 270 \text{ ps}$. Although the ICD lifetime is just $375 \text{ fs} / 0.3 = 1.25 \text{ ps}$, it is still much shorter than the radiative lifetime.”

- In a recent study, superradiant effects in resonant energy transfer between atoms, including the geometry dependence, have been investigated [arXiv:1912.05892]. In particular, the possibility for a complete suppression of the transfer rate inside a spherically symmetric distribution of donor atoms was found. The relation with this work needs to be discussed in the present manuscript.

We have not been aware of this study and discuss it now in the manuscript (see also reply to Reviewer #1).

At the end of the manuscript we added the following discussion: “As we mentioned in the introduction, we concentrate in this first study on the fundamental aspect of the impact of quantum light on ICD and do not discuss details concerning the choice of the cavity. To take the structure of the cavity into account, a QED approach analogous to that in Ref. [72] seems to be promising. Recently, this methodology was also used to investigate superradiant effects in resonant energy transfer in donor-acceptor ensembles and interesting dependence on geometry was found [73]. Such an approach also includes the effect of retardation not included in the present study. Retardation makes the impact of ICD more long range. However, for the transfer energies discussed in this work the impact of retardation is very small.”

- I wonder to which extent the predictions of the authors rely on the quantum description of the radiation field. It seems to me that similar - perhaps even identical - results would be obtained if the system of atoms was exposed to the coherent field of a classical electromagnetic (laser) wave, whose frequency is resonant to the atomic transition. Since this is a main point of the manuscript, I ask the authors to clearly indicate where their predictions based on quantum light differ from a semiclassical treatment.

We agree that, in principle, a collective initial excitation of the A atoms can be obtained in free space by a laser pulse that acts on all atoms with the same phase and amplitude. This excitation is related, but not the same as in a cavity. As seen, e.g., in Eq. (2), the collective excitation in the cavity is only a part of the polariton's wavefunction in the cavity. What is particularly important in the cavity is that the energy of the polariton states is well separated from the atomic energies and can thus be addressed directly by the laser exciting it. Consequently, a polariton state is much easier to selectively populate than a collective excitation of all atoms in a laser pulse. We agree that we have missed to discuss the relationship to the collective excitation by a laser, which we do now.

Apart from the accessibility of polaritons discussed above there is a particularly important difference between collective excitation in cavity, i.e., polaritons, and collective excitation by a laser. The coupling strength to the cavity determines the energy of the polaritons and this can be used to control ICD. Imagine an excited atom A^* whose excess energy (the energy difference between the excited and ground states) is somewhat smaller than the ionization energy of B and thus insufficient to ionize an atom B. As a consequence, a collective excitation of A atoms by a laser will not lead to ICD. This strongly

contrasts the potential a cavity has. Here, the energy of the upper polariton grows above the atomic excess energy as $N^{1/2}g$ and may exceed the ionization energy of B and ICD becomes operative. Another interesting scenario is met by choosing A^* and B such that the ICD channel is open in the absence of cavity and the lower polariton is populated. Then, as the energy decreases as $-N^{1/2}g$ below the atomic excess energy, one may by either increasing the number of atoms A or/and increasing the coupling strength g suddenly terminate the ICD process. Importantly, the cavity can also be used to manipulate the ICD rate in addition to switching on or off the ICD activity. The reason lies in the fact that the rate depends on the transferred energy and in cavity this energy can be varied by changing the energy of the polariton. It is worth mentioning that the rate is typically largest at the ICD threshold where the transferred energy equals the ionization energy of the neighbor.

We thank the Reviewer for his/her comment which has lead us to stress more clearly the usefulness of controlling ICD by having a cavity.

Several changes have been made to the manuscript.

At the end of the Abstract we have added the sentence: "It is stressed that in contrast to superposition states formed by a laser, forming polaritons by a cavity enables to control the emergence and suppression as well as the efficiency of ICD."

Above equation (1) the following sentence has been added: "Since the energies of polaritonic states can be manipulated, we shall see that cavities enable opening and closing the ICD channel which is not possible for a superposition state formed by a laser."

A new paragraph has been inserted and is now the first new paragraph on page 12: "There is a particularly important difference between the collective excitation in cavity, i.e., polaritons, and a collective excitation of the atoms formed by a laser. The coupling strength to the cavity determines the energy of the polaritons and this can be used to control ICD. Imagine an excited atom A^* whose excess energy (the energy difference between the excited and ground states) is somewhat smaller than the ionization energy of B and thus insufficient to ionize an atom B. As a consequence, a collective excitation of A atoms by a laser will not lead to ICD. This strongly contrasts the potential a cavity has. Here, the energy of the upper polariton grows above the atomic excess energy as $N^{1/2}g$ and may exceed the ionization energy of B and ICD becomes operative. As two rare gas examples we mention an ensemble of Ar atoms and Xe or Kr as a neighbor. For $Ar^*(3p \rightarrow 4s)$ the excess energy is 0.58 eV below the ionization energy of Xe, and for one of the $Ar^*(3p \rightarrow 3d)$ excited states it is just 0.02 eV below the ionization energy of Kr [62]. Another interesting scenario is met by choosing A^* and B such that the ICD channel is open in the absence of cavity and the lower polariton is populated. Then, as the energy decreases as $-N^{1/2}g$ below the atomic excess energy, one may by either increasing the number of atoms A or/and increasing the coupling strength g suddenly terminate the ICD process. Examples for this scenario could be an ensemble of Ne atoms and Ar as a neighbor like in the cluster discussed above, and an ensemble of Ar atoms and Kr as a neighbor. Here, the excess energy of $Ne^*(2s \rightarrow 3p)$ exceeds the ionization energy of Ar by 0.86 eV and that of $Ar^*(3p \rightarrow 5s)$ the ionization energy of Kr by just 0.07 eV [62]. Of course, other atoms and, in particular, molecules offer a larger variety for choosing candidates to open and close the ICD channel. Importantly, the cavity can also be used to manipulate the ICD rate in addition to switching on or off the ICD activity. The reason lies in the fact that the rate depends on the transferred energy and in cavity this energy can be varied by changing the energy of the polariton. It is worth mentioning that the rate is typically largest at the ICD threshold where the transferred energy

equals the ionization energy of the neighbor, see, e.g., [5,21,63]. ”

- In line with my previous remark, I do not quite agree with the statement on page 10 that “Without a cavity, a weak external laser would excite a single Ne atom”. In this situation the ensemble of N atoms A (after photoabsorption) would rather be in a coherent superposition state, as well. For the case N=2, corresponding expressions are given in the paper by Najjari, Muller and Voitkiv, New J. Phys. 14 (2012) 105028, which seems to be a follow-up on Ref.[44] with a detailed account of resonant photoionization in correlated three-atomic systems.

As we discussed above, we agree that a spatially and spectrally coherent laser field may also collectively excite the Ne atoms. But, this was not the situation we wanted to address here. We just wanted to present the numbers for the rates of a single Ne – Ar pair for the sake of later comparison. To avoid confusion, we made the following changes to the manuscript and added the reference mentioned by the Reviewer to the introduction (now [45]).

The new sentence now reads: “Without a cavity, when exciting a single Ne atom it will undergo ICD with the central Ar atom.”

The paper by Najjari, Muller and Voitkiv has been added, now Ref. [45].

- Minor point: The summation index in eqs.(2) and (3) should be i instead of n . The same holds for some subsequent sums in the following text.

We thank the Referee for pointing out these typos. They are now corrected.

Reviewers' Comments:

Reviewer #1:

Remarks to the Author:

The authors have responded to all the reviewers' comments. However, I remain unconvinced by their replies. There is still no meaningful analysis of the conditions under which a cavity could measurably affect ICD, and from my own back-of-the-envelope estimates, it seems unlikely that realistic cavities would have an effect, or at best have an effect that is not mentioned in the text (suppression of ICD due to the typically fast decay of polaritons).

-) Responding to the comment (by reviewer 3 and me) that there is very little difference between the discussed "cavity" excitation and simply coherent excitation of several atoms in free space, the authors now point out that polaritons have a different energy than bare atomic excited states, which could be used to modify ICD by opening/closing channels and modifying rates. This is correct, and I thank the authors for discussing it in detail now.

However, it is not true that a cavity is necessary to create "formally inaccessible" dark states, or that "many of the superposition states of free atoms formed by a laser other than the superradiant state are not dark". As mentioned in the previous report, already in free space the necessary sub-wavelength distance between atoms (as pointed out by the authors, ICD is negligible at distances of more than a few nm) means that only the superradiant state effectively couples with any free-space mode. This is due to their spatial dependence with $\exp(ikr) = \exp(2i\pi r/\lambda)$, which implies that any free-space photon couples to all atoms in the cluster with essentially the same phase (with variations on the order of $1/100$ radians), i.e., the superradiant state.

-) It is true that there are many articles in the literature that do not treat an explicit model for a cavity, and some of the criticisms I have raised certainly apply to them as well. I also agree with the authors that, in principle, "polaritonic states are formed in cavity and it is legitimate, relevant and timely to investigate ICD in these states". However, in the current case, the lack of any discussion of the possibly available parameter ranges is especially important, because the effects that are being neglected would give at least as big or even bigger effects than the ones considered here. First, as established above, the excitation of the superradiant state is simply a consequence of having a cluster of atoms at subwavelength distances (and is already discussed in the paper mentioned by reviewer three, which I was not previously aware of), and is thus not "the fundamental aspect of the impact of quantum light on ICD" that the authors claim it to be. Second, for a large ensemble, the rate is suppressed by a factor $1/N$, which rules out the use of "traditional" optical cavities with small single-atom coupling strengths if any significant change in energy due to strong coupling (i.e., polariton formation) is desired. Third, in subwavelength cavities, fast cavity mode losses (typically <10 fs) are unavoidable, which the polaritons will inherit. It appears that these will dominate compared to ICD rates. (It should be noted that the newly added analysis based on calculating free-space radiative decay rates completely neglects the effect of the cavity mode lifetime and is thus not relevant for these estimates.)

-) The authors now state that "As long as polariton states are formed and they are addressable, there is no need for reaching strong coupling". However, as polaritons are exactly the result of strong coupling, in its absence no polariton states are formed. It should be noted that since strong coupling depends on the coupling overcoming the losses, it is of course trivially present if losses are neglected in the model.

To summarize, I agree that the impact of cavities and polariton formation on ICD is an interesting topic in principle. However, in my opinion, for such a study to be suitable for publication in a high-impact journal such as Nature Communications, the topic should be covered in enough detail that a (somewhat) realistic estimate of the impact of cavities and their properties on the process can be obtained. In contrast, the aspect that is currently the sole focus of the manuscript is not even exclusive to cavities, and is even an unavoidable feature in free space (while subwavelength cavities

could in principle change this by allowing significant field variations on few-nm scales).

Reviewer #2:

Remarks to the Author:

The article is well readable and more clear thanks to the new parts introduced by the authors. The authors answered the question about the symmetry of other lattice-like system in an exhaustive way underlining how this first work can pave the way for the calculation of new geometrically interesting structures that may have an impact on the ICD phenomenon. All the issues that I referred in my comments, have been addressed by the authors and in my opinion I suggest the publication on Nature Communications.

Reviewer #3:

Remarks to the Author:

In the revised version, the authors have substantially improved their manuscript by addressing the remarks of the referees. The presentation of the physics is better balanced and more comprehensive now. In particular, the specific effect of a cavity mode on ICD - as compared with a classical laser field - has been explained. There is only one remaining point in this context:

- In the new paragraph on p.12 "There is a particularly important difference...", the authors argue that the product $\sqrt{N} \cdot g$ can be used to control and manipulate ICD in a cavity. They give detailed examples of atoms, including their energy levels. But they do not specify whether the product $\sqrt{N} \cdot g$ can attain values large enough to bridge the mentioned energy gaps which are as high as 0.86 eV in their examples. The authors should mention a concrete and realistic situation where a cavity has the desired properties in quantitative terms. This is necessary here to show the relevance of this important part of their argument, even though they do not discuss specific details of cavities in the rest of their manuscript.

With this final remark properly addressed, the manuscript deserves to be published.

Replies to Reviewers' comments

Reviewer #1 (Remarks to the Author):

-) Responding to the comment (by reviewer 3 and me) that there is very little difference between the discussed "cavity" excitation and simply coherent excitation of several atoms in free space, the authors now point out that polaritons have a different energy than bare atomic excited states, which could be used to modify ICD by opening/closing channels and modifying rates. This is correct, and I thank the authors for discussing it in detail now.

However, it is not true that a cavity is necessary to create "formally inaccessible" dark states, or that "many of the superposition states of free atoms formed by a laser other than the superradiant state are not dark". As mentioned in the previous report, already in free space the necessary sub-wavelength distance between atoms (as pointed out by the authors, ICD is negligible at distances of more than a few nm) means that only the superradiant state effectively couples with any free-space mode. This is due to their spatial dependence with $\exp(ikr) = \exp(2i\pi r/\lambda)$, which implies that any free-space photon couples to all atoms in the cluster with essentially the same phase (with variations on the order of $1/100$ radians), i.e., the superradiant state.

We are glad that the referee appreciates our main point concerning the important difference between superradiant states formed by a laser and polaritons formed in a cavity, namely that "polaritons have a different energy than bare atomic excited states" and this opens the way "to modify ICD by opening/closing channels and modifying rates." This is to our opinion an important finding that will be relevant not only to ICD but also to other kinds of interatomic (intermolecular) decay processes. We thank the referee again for forcing us in his first report to discuss the issue of the difference between polaritons and superradiant states.

We agree with the referee that a cavity is not necessary to create formally inaccessible dark states and we thank the referee for his/her explanation. This point is, however, not relevant to our work which is on the bright states, the polaritons, and does not appear in the manuscript.

-) It is true that there are many articles in the literature that do not treat an explicit model for a cavity, and some of the criticisms I have raised certainly apply to them as well. I also agree with the authors that, in principle, "polaritonic states are formed in cavity and it is legitimate, relevant and timely to investigate ICD in these states". However, in the current case, the lack of any discussion of the possibly available parameter ranges is especially important, because the effects that are being neglected would give at least as big or even bigger effects than the ones considered here. First, as established above, the excitation of the superradiant state is simply a consequence of having a cluster of atoms at subwavelength distances (and is already discussed in the paper mentioned by reviewer three, which I was not previously aware of), and is thus not "the fundamental aspect of the impact of quantum light on ICD" that the authors claim it to be. Second, for a large ensemble, the rate is suppressed by a factor $1/N$, which rules out the use of "traditional" optical cavities with small single-atom coupling strengths if any significant change in energy due to strong coupling (i.e., polariton formation) is desired. Third, in subwavelength cavities, fast cavity mode losses (typically <10 fs) are unavoidable, which the polaritons will inherit. It appears that these will dominate compared to ICD rates. (It should be noted that the newly added analysis based on calculating free-space radiative decay rates completely neglects the effect of the cavity mode lifetime and is thus not relevant for these estimates.)

The referee agrees with our point that "polaritonic states are formed in cavity and it is legitimate, relevant and timely to investigate ICD in these states" and adds "However, in the current case, the lack of any discussion of the possibly available parameter ranges is especially important ...". We agree that we have to address this issue and we do so now by adding text to the manuscript (see below). We would like to stress, however, that the development of new kinds of cavities is currently a fast growing field of research where increasing attention is paid to hybrid setups with the goal to combine enhancement of light-matter interaction with longer lifetimes.

The referee criticizes our sentence "the fundamental aspect of the impact of quantum light on ICD". As it stands this sentence might indeed be misleading. What we actually mean is not only the creation of a superposition of states, i.e., the polariton, but also the possibility to manipulate ICD in contrast to a superposition created by a laser. We have now revised this sentence making the point clear (see below). As we need strong coupling, "traditional" optical cavities are indeed unsuitable and one currently has to resort to sub-wavelength cavities like plasmonic nanocavities where single-molecule strong coupling has been achieved even at room temperature. As mentioned above, several promising developments of other methodologies have been reported with the aim of prolonging the cavity lifetime. Even with a cavity with short lifetime, ICD effects can be observed. Consider, for example, the two situations discussed in the manuscript: a) the ICD channel is energetically just open in free space and is closed in the lower polariton and b) the ICD channel is just closed in free space and opens in the upper polariton. In a) one will observe ICD in free space and not when populating the lower polariton and if sufficiently many atoms A surround the atom B, like in a ring, one can expect to see ICD by populating the upper polariton even if the cavity lifetime is short. In b) one does not observe ICD in free space and can observe ICD by populating the upper polariton. One should keep in mind that the theory is not limited to atoms and that molecules can have a much larger ICD rates and in some cases the ICD lifetime can be even below 10 fs (see, e.g., the review [21]). In addition, we stress that we have also addressed in the manuscript planar objects with many B neighbors (see Fig. 3b) and such objects enable faster ICD, in particular when molecules are involved.

We have substituted the sentence "There are several types of cavities available nowadays, but in this first study we concentrate on the fundamental aspect of the impact of quantum light on ICD and do not discuss details concerning the choice of the cavity." by the following text: "In this first study, we stress the fundamental aspect of the impact of quantum light on ICD, namely that in addition to forming superposition states it enables to control the emergence and suppression, as well as the efficiency of ICD. There are several types of cavities available nowadays, and this will be discussed too."

At the end of the manuscript, we also added the following clarifying text:

"Let us in the end touch upon the possibility to experimentally study the phenomena discussed in the present work. For this purpose, one will need set-ups in which strong-coupling regime with a small number of emitters can be achieved and kept long enough, that is, for times comparable with the ICD lifetime of the studied complex. Over the last decade we have witnessed a tremendous progress of quantum cavity technologies and various resonators have already been developed, ranging from dielectric cavities [80] and surface evanescent modes [81] to plasmonic cavities with "sub-wavelength" mode volumes [82] (for a recent review, see e.g. Ref. [62]). Strong-coupling regimes enabling creation of polaritonic states even with a single molecule at room temperature have already been reported [83]. Currently, these have been achieved using hybrid metallo-dielectric set-ups, in which a strongly sub-

wavelength cavity can be formed through localized surface plasmon resonances in nanometer gaps. The lifetimes of these resonances are, however, typically on the order of 10 fs, which puts some restrictions on the ICD or other energy-transfer processes that can be studied. There are ICD processes faster than 10 fs [21], but we note that this does not mean that the only ICD processes with lifetimes below 10 fs can be addressed. The effect of the cavity on the ICD process will be measurable even if the ICD lifetime is longer, because we can compare the ICD yield in free space and in the cavity. We note also that, as discussed above, through the energy split between the upper and lower polaritons one may open or close an ICD process. We mention that energy splits of about 0.4 eV have been reported [83] even in the single-molecule regime. Moreover, both a decrease of the ICD lifetime and an increase of the light-matter coupling can be achieved by designing appropriate 2D structures, similar to those shown in Fig. 3b, in particular when molecules are involved.

The fast development of quantum-resonator technologies may substantially enlarge the range of experimentally accessible ICD processes and systems in the near future. We hope that the present study will further motivate the research in this direction.”

The following references have also been added:

- [80] H. Deng et al., *Science* 298, 199 (2002).
- [81] P. M. Walker et al., *Nature Commun.* 6, 8317 (2015).
- [82] T. K. Hakala et al. *Phys. Rev. Lett.* 103, 053602 (2009).
- [62] A. Frisk Kockum et al. *Nat. Rev. Phys.* 1, 19 (2019).
- [83] R. Chikkaraddy et al., *Nature* 535, 127 (2016).

-) The authors now state that "As long as polariton states are formed and they are addressable, there is no need for reaching strong coupling". However, as polaritons are exactly the result of strong coupling, in its absence no polariton states are formed. It should be noted that since strong coupling depends on the coupling overcoming the losses, it is of course trivially present if losses are neglected in the model.

The cited sentence refers to basic theory and is in itself correct. However, it indeed may sound misleading when related to practically available cavities which do have sizeable losses. Due to these losses, what is called strong coupling is needed in order to form the polariton. To remedy the situation, we reformulated the sentence and added clarifying text.

We replaced the last sentence in the discussion of the ICD in the ArNe₁₂ cluster by the following text: "We mention here that as long as polaritonic states are formed, the above findings on the ICD rate are independent on the coupling strength g . However, available cavities do have sizeable losses and strong coupling is needed in order to form the polariton [62]. See also the discussion at the end. We also note that there are experimental investigations of ICD in much larger clusters, for instance, ICD in Ne clusters with about 5000 atoms [63], ICD in doped He nanodroplets with about 50000 atoms [64], ICD in mixed NeKr clusters with about 1000 atoms [65] and ICD in He nanodroplets with about 10000 and 50000 atoms [66,67]."

The following references have also been added:

- [62] A. Frisk Kockum et al. *Nat. Rev. Phys.* 1, 19 (2019).
- [63] K. Nagaya et al., *Nature Commun.* 7, 13477 (2016).

- [64] A. C. LaForge et al., Nat. Phys. 15, 247 (2019).
[65] T. Arion et al., JCP 134, 074306 (2011).
[66] Y. Ovcharenko et al., PRL 121, 073401 (2014).
[67] A.C. LaForge et al , PRX 11, 021011 (2021).

To summarize, I agree that the impact of cavities and polariton formation on ICD is an interesting topic in principle. However, in my opinion, for such a study to be suitable for publication in a high-impact journal such as Nature Communications, the topic should be covered in enough detail that a (somewhat) realistic estimate of the impact of cavities and their properties on the process can be obtained. In contrast, the aspect that is currently the sole focus of the manuscript is not even exclusive to cavities, and is even an unavoidable feature in free space (while subwavelength cavities could in principle change this by allowing significant field variations on few-nm scales).

The major differences between ICD of superposition states created in free space and by cavities are clearly discussed in the manuscript. The fact that the energy of the polariton differs from that of the excitation of the emitter in free space opens the door to enable, suppress and change the efficiency of ICD by a cavity. The referee criticizes “the lack of any discussion of the possibly available parameter ranges”. We discussed the present situation in the development of cavities above and added new text to the manuscript.

Reviewer #3 (Remarks to the Author):

In the revised version, the authors have substantially improved their manuscript by addressing the remarks of the referees. The presentation of the physics is better balanced and more comprehensive now. In particular, the specific effect of a cavity mode on ICD - as compared with a classical laser field - has been explained. There is only one remaining point in this context:

- In the new paragraph on p.12 "There is a particularly important difference...", the authors argue that the product $\sqrt{N} \cdot g$ can be used to control and manipulate ICD in a cavity. They give detailed examples of atoms, including their energy levels. But they do not specify whether the product $\sqrt{N} \cdot g$ can attain values large enough to bridge the mentioned energy gaps which are as high as 0.86 eV in their examples. The authors should mention a concrete and realistic situation where a cavity has the desired properties in quantitative terms. This is necessary here to show the relevance of this important part of their argument, even though they do not discuss specific details of cavities in the rest of their manuscript.

With this final remark properly addressed, the manuscript deserves to be published.

In each example we gave a large and a small energy gap. This was done to demonstrate the range of possibilities present even in simple, i.e., rare gas, atoms. Due to the remark of the referee, we notice that the large number for the energy gap is somewhat misleading in connection to cavities. We have added a clarifying sentence (see below) and thank the referee for his/her remark.

The following sentence has been added: "The above mentioned larger energy gaps of 0.58 and 0.86 eV cannot be overcome by the current available cavity technology, see the discussion below. They are merely given to demonstrate the range of possibilities present even in simple, i.e., rare gas, atoms."

In addition, we also added an explanatory text discussing cavities and their properties:

"Let us in the end touch upon the possibility to experimentally study the phenomena discussed in the present work. For this purpose, one will need set-ups in which strong-coupling regime with a small number of emitters can be achieved and kept long enough, that is, for times comparable with the ICD lifetime of the studied complex. Over the last decade we have witnessed a tremendous progress of quantum cavity technologies and various resonators have already been developed, ranging from dielectric cavities [80] and surface evanescent modes [81] to plasmonic cavities with "sub-wavelength" mode volumes [82] (for a recent review, see e.g. Ref. [62]). Strong-coupling regimes enabling creation of polaritonic states even with a single molecule at room temperature have already been reported [83]. Currently, these have been achieved using hybrid metallo-dielectric set-ups, in which a strongly sub-wavelength cavity can be formed through localized surface plasmon resonances in nanometer gaps. The lifetimes of these resonances are, however, typically on the order of 10 fs, which puts some restrictions on the ICD or other energy-transfer processes that can be studied. There are ICD processes faster than 10 fs [21], but we note that this does not mean that the only ICD processes with lifetimes below 10 fs can be addressed. The effect of the cavity on the ICD process will be measurable even if the ICD lifetime is longer, because we can compare the ICD yield in free space and in the cavity. We note also that, as discussed above, through the energy split between the upper and lower polaritons one may open or close an ICD process. We mention that energy splits of about 0.4 eV have been reported [83] even in the single-molecule regime. Moreover, both a decrease of the ICD lifetime and an increase of the light-

matter coupling can be achieved by designing appropriate 2D structures, similar to those shown in Fig. 3b, in particular when molecules are involved.

The fast development of quantum-resonator technologies may substantially enlarge the range of experimentally accessible ICD processes and systems in the near future. We hope that the present study will further motivate the research in this direction.”

The following references have also been added:

- [80] H. Deng et al., *Science* 298, 199 (2002).
- [81] P. M. Walker et al., *Nature Commun.* 6, 8317 (2015).
- [82] T. K. Hakala et al. *Phys. Rev. Lett.* 103, 053602 (2009).
- [62] A. Frisk Kockum et al. *Nat. Rev. Phys.* 1, 19 (2019).
- [83] R. Chikkaraddy et al., *Nature* 535, 127 (2016).

Reviewers' Comments:

Reviewer #1:

Remarks to the Author:

The authors have now given additional details about possible cavity architectures and available parameter ranges, and have convinced me that there is at least a chance that the effects they discuss could be observable. I am thus happy to recommend publication.

Reviewer #3:

Remarks to the Author:

In the second revised version of their manuscript the authors have addressed my previous remark in a satisfactory way. The discussion has now been completed by drawing connections between the parameters of the theory and the properties of experimentally available cavities. This amendment was important in my opinion in order to orient readers about the requirements for experimental verification of the interesting effects predicted by the authors.

I recommend the manuscript in its current form for acceptance.